# Excess cost of care associated with sepsis in cancer patients: Results from a population-based case-control matched cohort

**Michelle Tew**[1,2,3]*, **Kim Dalziel**[1], **Karin Thursky**[2,3,4], **Murray Krahn**[5],
**Lusine Abrahamyan**[5], **Andrew M. Morris**[6], **Philip Clarke**[1,7]

**1** Centre for Health Policy, Melbourne School of Population and Global Health, University of Melbourne, Melbourne, Australia, **2** National Centre for Infections in Cancer, Peter MacCallum Cancer Centre, Melbourne, Australia, **3** Department of Health Services Research, Peter MacCallum Cancer Centre, Melbourne, Australia, **4** National Centre for Antimicrobial Stewardship, Royal Melbourne Hospital, Melbourne, Australia, **5** Toronto Health Economics and Technology Assessment Collaborative, Toronto General Hospital Research Institute, University of Toronto, Toronto, Canada, **6** Department of Medicine, Division of Infectious Disease, Sinai Health, University Health Network and University of Toronto, Toronto, Canada, **7** Health Economics Research Centre, Nuffield Department of Population Health, University of Oxford, Oxford, United Kingdom

* michelle.tew@unimelb.edu.au

**Data Availability Statement:** The data set from this study is held securely in coded form at ICES. While data sharing agreements prohibit ICES from

## Abstract

### Background

Cancer patients are at significant risk of developing sepsis due to underlying malignancy and necessary treatments. Little is known about the economic burden of sepsis in this high-risk population. We estimate the short- and long-term healthcare costs of care of cancer patients with and without sepsis using individual-level linked-administrative data.

### Methods

We conducted a population-based matched cohort study of cancer patients aged ≥18, diagnosed between 2010 and 2017. Cases were identified if diagnosed with sepsis during the study period, and were matched 1:1 by age, sex, cancer type and other variables to controls without sepsis. Mean costs (2018 Canadian dollars) for patients with and without sepsis up to 5 years were estimated adjusted using survival probabilities at partitioned intervals. We estimated excess cost associated with sepsis presented as a cost difference between the two cohorts. Haematological and solid cancers were analysed separately.

### Results

77,483 cancer patients with sepsis were identified and matched. 64.3% of the cohort were aged ≥65, 46.3% female and 17.8% with haematological malignancies. Among solid tumour patients, the excess cost of care among patients who developed sepsis was $29,081 (95% CI, $28,404-$29,757) in the first year, rising to $60,714 (95%CI, $59,729-$61,698) over 5 years. This was higher for haematology patients; $46,154 (95%CI, $45,505-$46,804) in year 1, increasing to $75,931 (95%CI, $74,895-$76,968).

making the data set publicly available, access may be granted to those who meet pre-specified criteria for confidential access, available at https://www.ices.on.ca/DAS. The full data set creation plan and underlying analytic code are available from the authors upon request, understanding that the programs may rely upon coding templates or macros that are unique to ICES.

**Funding:** Michelle Tew is jointly supported by the NHMRC funded Centre for Research Excellence in Total Joint Replacement (1116325) and Centre for Improving Cancer Outcomes Through Enhanced Infection Services (1116876), Melbourne Research Scholarship and Australian Research Council Centre of Excellence in Population Ageing Research (CEPAR). Murray Krahn is supported by a Tier 1 Canada Research Chair in Health Technology Assessment. The funders had no role in study design, data collection and analysis, decision to publish, or preparation of the manuscript.

**Competing interests:** The authors have declared that no competing interests exist.

## Conclusions

Sepsis imposes substantial economic burden and can result in a doubling of cancer care costs, particularly during the first year of cancer diagnosis. These estimates are helpful in improving our understanding of burden of sepsis along the cancer pathway and to deploy targeted strategies to alleviate this burden.

## Introduction

Sepsis is a potentially life-threatening organ dysfunction caused by the body's response to infection [1]. It is a major cause of morbidity and mortality [2–7] contributing up to one-fifth of deaths reported globally in 2017 [8]. Patients with cancer are at high risk of developing sepsis. It is estimated that cancer patients are 10-times more likely to develop sepsis compared to non-cancer patients [9]. Numerous factors contribute to this risk including underlying malignancy, immune dysfunction following life-saving treatments, recurrent hospitalisations, and the need for invasive procedures. The cost of managing sepsis is high. Sepsis is among the most expensive conditions treated in hospitals, amounting to approximately $24 billion in hospital costs in the US in 2013 alone [10, 11]. This tops other high-cost hospitalisations such as acute myocardial infarctions ($12.1 billion). Based on US projections, the burden of cancer is even larger at $158 billion [12]. While much is known about cancer care costs at various phases of patient's cancer journey from initial diagnosis to end-of-life, it is unclear how much of this burden is attributed to sepsis.

Although sepsis incidence and its associated outcomes such as mortality have been well described in the literature [5–9, 13–16], the majority of these studies were focused on severe sepsis and were not specific to cancer. Among those that quantified costs, estimates [7, 14–17] have relied on hospital admissions data and showed that severe sepsis cancer hospitalisations can cost more than three times as much as non-severe cancer hospitalisations [14]. Hospitalisation data is likely to capture only the most severe cases and potentially miss sepsis burden incurred outside of the hospital. Previous studies have shown that the prevalence of less severe forms of sepsis is much higher than severe sepsis or septic shock, consequently the overall disease burden of sepsis is expected to be much larger [7, 18]. The burden of sepsis is also likely to extend beyond the index hospitalisation as growing evidence suggests that sepsis increases the risk of rehospitalisation [19, 20], cognitive decline [4, 21] cardiovascular complications [22, 23] and death [23–25] in studies assessing longer-term outcomes. Limited attention has focused on the economic burden of sepsis in the high-risk cancer population. An understanding of the cost burden of sepsis beyond acute hospital care is needed to enable healthcare providers and policy makers to develop strategies for more efficient care. Currently, robust long-term cost estimates that adequately capture this in cancer populations are lacking.

In this study, we aim to describe short- and long-term healthcare costs of care of adult cancer patients with and without sepsis in Ontario, Canada. We use population-linked administrative data to capture health services use including those beyond inpatient hospitalisations. This provides a unique opportunity to study the economic burden of sepsis across the entire health care system and across the cancer care continuum. These cost estimates will provide an indicator on the magnitude of sepsis burden on top of cancer care and can be helpful to health administrators and policy makers in aligning appropriate resources for health workforce capacity, infrastructure including sepsis programs to achieve efficient allocation of public resources across various services and inform on need for further research.

## Methods

We conducted a population-based retrospective cohort study using patient-level administrative health data to determine healthcare costs among adult patients who developed sepsis compared to those without sepsis up to 5 years following cancer diagnosis. This study protocol was approved by research ethics board at the University of Toronto (#37526) and University of Melbourne (#1953663).

### Patient cohort and data source

Patients were selected from the Ontario Cancer Registry [26] and included in study if aged 18 and above, whose first diagnosis for a primary cancer occurred between January 1, 2010 and December 31, 2017. Patients were followed until death or end of analysis period, March 31, 2018. Patients were excluded if cancer diagnosis was first identified at death, or if there was previous cancer diagnosis prior to the study period. Cancer patients were classified by tumour site according to International Classification of Diseases-Oncology (ICD-O) topography code corresponding to their primary cancer diagnosis and classified into two broad groups—haematological and solid cancers [27].

Individual-level data on all patient healthcare resource use from diagnosis up to study end date were obtained from ICES in Toronto, Ontario. These data describe resource utilisation for residents of Ontario, Canada (population 14.6 million) covered by Ontario Health Insurance Plan (97%). The data sources include inpatient hospitalisations, emergency department, cancer clinic visits, physician services, diagnostic tests, long-term care, prescription drugs, chemotherapy and radiotherapy (S1 Appendix for details). These datasets were linked using unique encoded identifiers and analysed at ICES. These data sources capture up to 90% of all healthcare resources provided universally and paid for by Ontario Ministry of Health and Long-Term Care [28]. Healthcare services and cost relating to community services, outpatient prescriptions for those aged 65 and below (and not receiving social assistance) and other healthcare costs paid out-of-pocket are not captured. Despite this, these data sources represent the best available and have been used in numerous other costing analyses [29–31].

### Identification of sepsis, cases and controls

Sepsis is defined as life-threatening organ dysfunction caused by dysregulated host response to infection [1] and was identified using ICD-10-CA diagnosis codes captured within the data source. We applied the explicit and implicit definition for case finding recently published by the Global Burden of Disease Group [8] which reflects the most current definition of sepsis, and thus allowed for better case ascertainment (S2 Appendix). Cancer patients were classified as cases if identified with sepsis within the 5-year study period and within 1 month prior to cancer diagnosis. The '1 month prior' inclusion period allowed for some flexibility in accuracy of diagnosis dates and also inclusion of patients whose sepsis presentation may have been the result of undiagnosed cancer [32]. Cancer patients were classified as potential controls if no sepsis record was identified throughout the study period. Cases (cancer patients with sepsis) were hard (exact) matched 1:1 by age (+/-2 years), sex, cancer type, year of cancer diagnosis and rurality to cancer patients without sepsis (controls) selected from the same patient cohort [33, 34].

### Estimating costs

The cost analysis is undertaken from the healthcare payer perspective. Costs for all healthcare services were estimated as described in [28]. Costs for inpatient hospitalisations, emergency

department and ambulatory care visits and long-term care were estimated by multiplying resource intensity weight (a measure of resource utilisation) by cost per weighted case or day. Costs for medications, chemotherapy and physician services were available directly in the data. Radiation costs were based on the intensity of resource use captured by National Hospital Productivity Improvement Program (NHPIP) codes and unit cost obtained from Earle et al. [35]. A detailed costing methodology is described in S1 Appendix. All costs were adjusted to 2018 Canadian dollars using healthcare component of the Statistics Canada Consumer Price Index [36].

As patients were observed over different time periods, not all patients had complete cost information across the entire 5-year period. This meant that for these patients, a portion of the relevant healthcare costs was unobserved because their observation period ended prematurely (right censored). Therefore, to estimate costs with incomplete follow-up data (common in longitudinal studies), methods that take into account this form of censoring are required to ensure unbiased cost estimates [37, 38]. This was done by partitioning the study period into monthly intervals and adjusting observed costs at each interval by the survival probability of corresponding interval [39]. This approach was chosen because it compared well to other approaches such as the inverse probability weighted (IPW) estimator approach, particularly with smaller time intervals as we have employed in our analysis and in the presence of heavy censoring, simple IPW methods may produce unstable estimates [38, 40]. This provided estimates for mean monthly cost of care for cancer patients with sepsis (cases) and without sepsis (controls). The average total (cumulative) cost across 5 years was estimated as the sum across 60-monthly intervals. We employed the established "excess" or "net" cost approach to obtain costs attributable to sepsis, where estimated healthcare costs of cancer patients without sepsis was subtracted from the cost of cancer patients who developed sepsis [41, 42]. As it is often difficult to separate specific costs that are sepsis- or cancer-related, this approach has been applied in numerous other costing analyses to describe economic burden associated with cancer [29, 30, 41–43].

As costs and survival probabilities are likely to be different between haematological and solid cancers, these patients were analysed separately. As cost of care at the end-of-life which is expected to be high [29, 42] and an important contributor to overall costs, costs in the last 6 months of life were segmented into a separate category of 'terminal care costs' to distinguish these. Sub-group analyses by sex and age groups were also conducted. Bootstrapping with 1000 replicates was used to calculate the 95% confidence intervals for all costs. All tests of significance used two-sided P-values at less than 0.05. Analyses were conducted using Stata version 16.

### Sensitivity analysis

A number of additional analyses were performed to test the robustness of the results. We tested the sensitivity of our results by (i) including more matching variables such as income quintiles and summary scores of other socioeconomic measures; e.g. dependency, deprivation, ethnic concentration (at the expense of identifying suitable controls), (ii) excluding the 1-month pre-diagnosis period from our sepsis case definition, (iii) alternate case definition of sepsis (Sepsis-2) [44] and (iv) duration attributed to end-of-life costs (12 months rather than 6 as in our main analysis).

## Results

### Study cohort and patient characteristics

A total of 485,105 cancer patients met eligibility criteria of the study and 83,028 patients (17.1%) experienced at least one sepsis episode over study period. Of these cases, matches were

found for 77,483 (93.3%) patients. 64.3% were aged 65 and above, 46.3% were female and 17.8% had haematological malignancies. Among those with solid tumours, lung (18.2%), colorectal (16.3%), breast (9.8%) and prostate (8.7%) were the most common cancer types. Leukemia (59.4%) formed the largest proportion of patients in the haematology group. Table 1 describes baseline characteristics of cancer patients with sepsis by malignancy type.

Across the 5-year period, a large proportion of sepsis episodes occurred in the first year of cancer diagnosis. Among haematology patients, 68.2% of first sepsis episodes were within the first year and this was 66.3% for solid tumour patients (S3 Appendix). The median time from cancer diagnosis to the first sepsis episode was 3 months (IQR, 0–12) for haematology patients and 4 months (IQR, 0–16) for solid tumour patients. A higher proportion of haematology patients (41.0%) had >1 episode of sepsis compared to solid tumour patients (26.7%). The difference in five-year overall survival between cancer patients with sepsis and without sepsis was statistically significant (log rank test p<0.001) across both cancer types (S4 Appendix).

Overall, controls were well matched to cases, except on income quintiles (S5 Appendix). Unmatched individuals were observed to be older, more likely to be male, have a haematological malignancy and more likely to have died by the end of the study period (S6 Appendix).

## Cost of care of sepsis

The monthly cost of care by malignancy type across the 5-year period for sepsis cancer patients and matched controls are presented in Fig 1. In general, healthcare costs were higher among those with sepsis compared to those without sepsis irrespective of malignancy types. Cost of care of sepsis for haematology patients is at least double that of a non-sepsis patient, and this difference is greatest particularly in the first 12 months of cancer diagnosis. In solid tumour patients, sepsis resulted in at least a 61% increase in overall cost of care. Across the 5-year period, total excess (net) cost of care among patients who developed sepsis is substantial (Table 2) and is higher among haematology patients at $75,931 (95% CI, 74,895–76,968) compared to solid tumour patients at $60,714 (95% CI, 59,729–61,698).

A large proportion of excess cost of care among patients who developed sepsis was incurred in the first 12 months of cancer diagnosis and this gradually declined in subsequent months (Fig A2 in S7 Appendix). Across the 5-year period, approximately 39% of the total excess cost was attributed to terminal care cost (last 6 months of life) in solid tumour patients. In haematology patients, the proportion of terminal care cost increased gradually over the 5-year period, from 36.8% at six months to above 90% by year 5.

Fig 2 shows variations in 1-year cumulative excess sepsis cost across different sub-groups by sex and age categories. Similar patterns were observed for costs over a longer time horizon (2- and 5-years). Costs of care and the resulting excess cost among patients who developed sepsis were higher for males and highest among males with a haematological malignancy. Across age groups, costs of care generally rose with increasing age. Among those aged ≤65, 5-year healthcare costs of patients with sepsis were at least twice that compared to patients without sepsis, resulting in higher excess cost among these patients compared to older patients. These results indicate that the burden of sepsis was highest among those in younger age categories (full results in S8 Appendix).

## Sensitivity analysis

The inclusion of additional matching variables and exclusion of the 1-month pre-diagnosis period from our sepsis case definition did not substantially change our cost estimates (variations between -3% and 8%). Cost estimates were sensitive to the sepsis definitions used. Using the Sepsis-2 definition reduced estimated excess cost, 14–33% lower costs for solid tumours

**Table 1. Characteristics of cancer patients with sepsis by malignancy type.**

| Characteristic | Haematology (n = 13,762) | Solid tumour (n = 63,721) |
|---|---|---|
| Age, No. (%) | | |
| 18–34 | 496 (3.6) | 964 (1.5) |
| 35–44 | 554 (4.0) | 1,962 (3.1) |
| 45–54 | 1,343 (9.8) | 6,141 (9.6) |
| 55–64 | 2,541 (18.5) | 13,655 (21.4) |
| 65–74 | 3,486 (25.3) | 18,639 (29.3) |
| 75–84 | 3,521 (25.6) | 15,821 (24.8) |
| 85+ | 1,821 (13.2) | 6,539 (10.3) |
| Female, No. (%) | 6,115 (44.4) | 29,765 (46.7) |
| Urban/rural residence, No. (%) | | |
| Urban | 12,236 (88.9) | 56,034 (88.3) |
| Rural | 1,526 (11.1) | 7,473 (11.7) |
| Income quintile, No. (%) | | |
| Low | 2,878 (21.0) | 14,509 (22.8) |
| Medium-low | 2,955 (21.5) | 13,788 (21.7) |
| Medium | 2,679 (19.5) | 12,531 (19.7) |
| Medium-high | 2,628 (19.1) | 11,675 (18.4) |
| High | 2,590 (18.9) | 11,060 (17.4) |
| Type of cancer, No. (%) | | |
| *Haematology* | | |
| Leukaemia | 8,174 (59.4) | - |
| Lymphoma | 3,367 (24.5) | - |
| Myeloma | 2,221 (16.1) | - |
| *Solid tumour* | | |
| Lung | - | 11,601 (18.2) |
| Colorectal | - | 10,415 (16.3) |
| Breast [a] | - | 6,271 (9.8) |
| Prostate | - | 5,565 (8.7) |
| Bladder | - | 2,929 (4.6) |
| Pancreatic | | 2,627 (4.1) |
| Stomach | | 2,224 (3.5) |
| Head and neck | | 2,220 (3.5) |
| Kidney | | 1,960 (3.1) |
| Liver | | 1,916 (3.0) |
| Melanoma | | 1,812 (2.8) |
| Others | - | 14,181 (22.3) |
| Year of cancer diagnosis, No. (%) | | |
| 2010 | 1,767 (12.8) | 7,881 (12.4) |
| 2011 | 1,698 (12.3) | 8,441 (13.3) |
| 2012 | 1,725 (12.5) | 8,670 (13.6) |
| 2013 | 1,772 (12.9) | 8,925 (14.0) |
| 2014 | 1,799 (13.1) | 8,524 (13.4) |
| 2015 | 1,855 (13.5) | 8,145 (12.8) |
| 2016 | 1,694 (12.3) | 7,571 (11.9) |
| 2017 | 1,452 (10.6) | 5,564 (8.7) |

[a] Breast cancer among females.

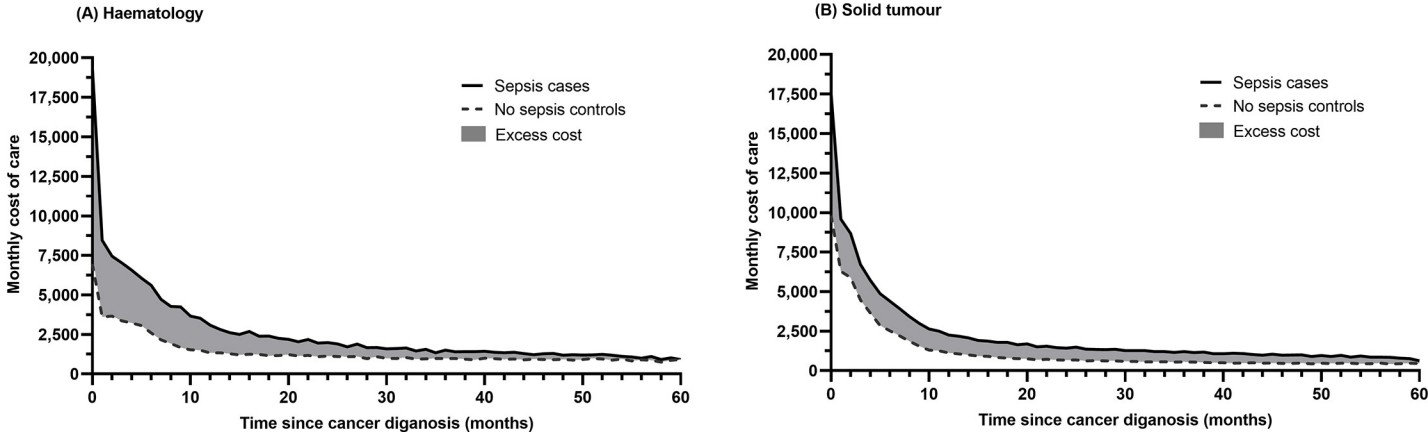

**Fig 1. Mean monthly cost of care by malignancy type.** (A) Haematolgy. (B) Solid tumour. The shaded area represents excess cost associated with sepsis, solid line represents mean monthly cost of care among sepsis (cases) and dotted line for no sepsis (controls).

and 3–13% lower costs for haematology. Unsurprisingly, the proportion of end-of-life cost increased from 57% to 77% for haematology patients and from 39% to 54% for solid tumour patients as the length of terminal care increased from 6 to 12 months. Full results are presented in S9 Appendix.

## Discussion

This study used patient-level administrative data to estimate for the first time whole of system healthcare cost of cancer patients with and without sepsis and has documented the excess cost of care associated with sepsis over a 5-year period. The cost of care of cancer patients who developed sepsis is substantial–up to 90% higher compared to patients without sepsis. This translated into an excess cost associated with sepsis of $29,081 in the first year, rising to $60,714 over 5 years for solid malignancies. This was higher for haematology; $46,154 in the first year, increasing to $75,931 after 5 years. These findings indicate that sepsis is a high cost, high mortality condition in cancer patients requiring urgent need for interventions and health policies to alleviate this significant burden.

**Table 2. Cumulative cost of care ($CAD 2018, 95% CI) between sepsis cases and matched controls.**

| Time since cancer diagnosis (months) | Haematology | | | Solid tumour | | |
|---|---|---|---|---|---|---|
| | Sepsis cases | Matched controls (no sepsis) | Excess cost | Sepsis cases | Matched controls (no sepsis) | Excess cost |
| 1 | 19,520 (19,174–19,867) | 7,026 (6,859–7,193) | 12,494 (12,105–12,883) | 17,403 (17,069–17,737) | 9,765 (9,606–9,925) | 7,638 (7,272–8,004) |
| 3 | 35,270 (34,866–35,675) | 14,255 (14,050–14,459) | 21,016 (20,562–21,470) | 35,592 (35,180–36,005) | 22,008 (21,767–22,249) | 13,585 (13,107–14,062) |
| 6 | 55,155 (54,661–55,650) | 23,731 (23,484–23,977) | 31,425 (30,884–31,966) | 53,064 (52,562–53,566) | 33,038 (32,749–33,326) | 20,026 (19,449–20,603) |
| 12 | 81,316 (80,718–81,915) | 35,162 (34,857–35,467) | 46,154 (45,050–46,804) | 72,817 (72,230–73,405) | 43,736 (43,400–44,073) | 29,081 (28,404–29,757) |
| 24 | 110,328 (109,624–111,032) | 49,773 (49,410–50,136) | 60,555 (59,786–61,323) | 94,456 (93,787–95,124) | 54,174 (53,793–54,554) | 40,282 (39,496–41,068) |
| 60 | 160,109 (159,204–161,014) | 84,178 (83,626–84,730) | 75,931 (74,895–76,968) | 133,683 (132,842–134,524) | 72,969 (72,498–73,440) | 60,714 (59,729–61,698) |

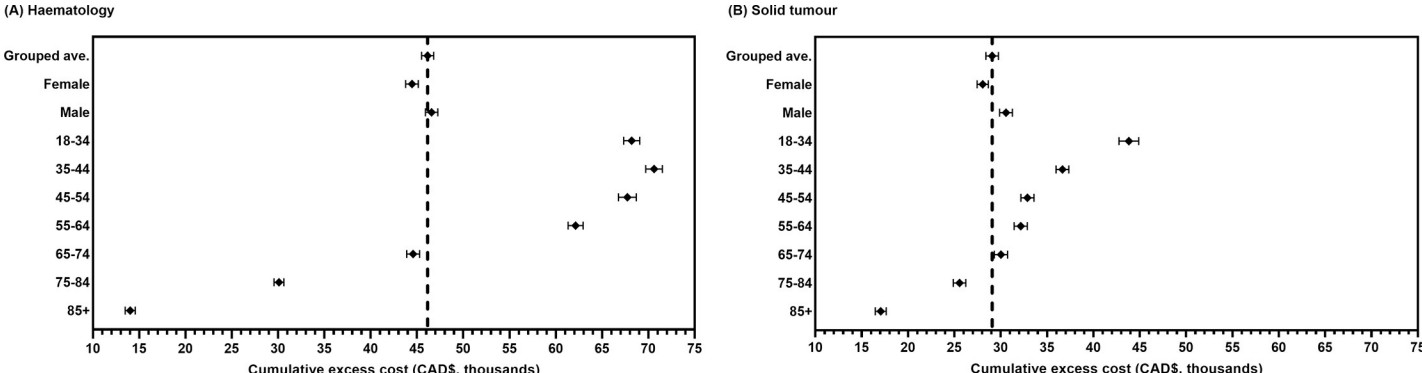

**Fig 2. Variations in the 1-year cumulative excess cost by sex and age groups.** (A) Haematolgy. (B) Solid tumour. The dotted vertical line represents the excess cost presented in our main analysis (overall grouped average). Error bars represent the 95% confidence intervals.

Excess cost among patients who developed sepsis was highest in the first month of cancer diagnosis and remained high through the first year (contributing to approximately two-thirds of sepsis episodes observed). This is likely to coincide with the initial phase of cancer care covering the diagnosis and initial treatment (chemotherapy or surgical intervention or radiotherapy) when patients are at increased risk of sepsis with neutropenic fever and other infections. This highlights the need for increased attention at this critical stage on the cancer pathway. Sepsis has been found to commonly occur within 14 days of cancer treatment [32], therefore strategies to increase vigilance and improve early recognition and timely interventions may be warranted in helping reduce this significant excess burden. Numerous intervention strategies and clinical pathways have been developed to assist clinicians in early identification of sepsis and prompt initiation of appropriate treatments to improve outcomes of patients with sepsis [45]. Clinical pathways for sepsis that provide protocolised management approaches such as sepsis bundles from the Surviving Sepsis Campaign have demonstrated effectiveness in reducing mortality by up to 50% [45–47]. However, majority of currently available literature are not specific to high-risk populations such as cancer. A clinical pathway with a whole-of-systems approach has the potential to alleviate the burden and costs of sepsis in cancer patients [17]. The findings from this analysis indicate the need to strengthen such initiatives for prompt sepsis identification and treatment particularly in the first year of cancer diagnosis and in managing (and preventing) subsequent sepsis episodes. Effective implementation of such strategies can have a big impact on improving outcomes of cancer patients with sepsis and in driving down the excess cost burden of sepsis as well as the future cost of managing sepsis and cancer.

The excess cost burden of sepsis was found be highest among haematological malignancies, males and younger (below age 55 years) patients. The higher cost of care among haematology patients compared to solid tumour patients was unsurprising as similar findings have been reported [12, 29, 43] reflecting more intensive chemotherapy regimens that may then progress to allogenic stem cell transplant within the first few months of diagnosis which may predispose patients to an increased risk of developing sepsis [14]. We had anticipated excess cost of sepsis to remain substantial over the study period due to morbidities related to sepsis [4, 48, 49] and increased risk of sepsis in cancer survivors [50] which necessitates a greater level of care. However, we observed a long tail with much lower excess cost (Fig 1) over the 5-year time horizon. This could reflect the acute nature of sepsis, which is treated episodically, requiring intensive and expensive treatments when it occurs (likely within the inpatient setting where healthcare costs are high). Similar tapering trends in cost in the months following the initial diagnosis period have also been observed in previous studies similar that observed in the current study

which may reflect the end of the intensive treatment and follow-up period [29, 43, 51]. It could also be due to a multitude of other factors; for instance, episodes of sepsis can lead to changes in the management of these patients including reduced intensity of treatments, cessation of therapy and/or prevention strategies for further episodes [52, 53]. Further research to better understand pathways of care of cancer patients with sepsis is warranted. Enhancing our understanding of the role of different healthcare services can help guide policy design and allocation of healthcare sources to alleviate both the cost and illness burden of sepsis on health system as well as patients.

A key strength of this study is the use of population-linked healthcare datasets which captures nearly all publicly funded healthcare services thus providing a whole of system view of the impact of sepsis. It provides a valuable opportunity to gain critical insights on the implications and burden of sepsis across the cancer care continuum which was not possible without access to robust linked-administrative datasets and systems. Data generated from contact with the healthcare system provides important real-world evidence and a more accurate reflection of the economic burden across the healthcare system. They provide a broader and longer view of the impact of sepsis in cancer patients, going beyond the limited hospital estimates currently available. These cost estimates are helpful in informing resource allocation and health policy prioritisation considerations and can also be used in cost-effectiveness models for decisions on sepsis interventions. They are useful in helping inform development of sepsis programs and policies across the cancer care continuum, which can include prevention, screening, treatment and end-of-life care.

With the growing use of novel cancer treatment strategies such as immunotherapies as emerging standards of care, this could change patterns of sepsis currently observed [54, 55]. In light of this, cost estimates presented in this study can be an important input for economic models when evaluating the value of these expensive new therapies and inform policy decisions on the value of cancer care. The large differences in costs of care between haematology and solid tumour patients requires further examination into the impact of sepsis across different tumour types, particularly haematological malignancies. For example, patients with acute myeloid leukemia tend to have poorer outcomes and may be more susceptible to sepsis. Additionally, future research should also aim to better understand how the duration, timing and severity of sepsis will impact costs and this can contribute towards a fuller understanding of the economic burden of sepsis in cancer patients.

There is a lot of heterogeneity in capturing sepsis from administrative datasets which can lead to variations in our understanding and monitoring of sepsis [56]. This can also result in differences in cost estimates produced as demonstrated in our sensitivity analysis (S9 Appendix). Applying an alternate sepsis definition (Sepsis-2) resulted in more sepsis cases captured which produced lower cost estimates. This may be due to the high negative predictive value of the approach (i.e. potential of increase in false positives) [44]. In the current analysis, we applied a comprehensive approach reflecting the most recent sepsis definition to ensure better case ascertainment [1, 8]. Further, capturing sepsis cases using the explicit and implicit codes provides a more realistic capture of sepsis and its associated costs than would be reflected through sepsis-specific codes only [57].

It is acknowledged that health care costs can vary across jurisdictions, particularly among those with differently funded health systems; for instance, cancer care costs often higher in the US compared to universal, publicly funded health systems in Canada and New Zealand [12, 29, 30, 42, 43, 51]. However, given the similarity in disease patterns and cancer care strategies across the developed world, these results may be generalisable and can be valuable to other similar settings that currently lack a clear view of the economic burden of sepsis in cancer patients. Similar studies using large population-based samples for generating real-world

estimates will be helpful in enhancing our understanding of the role of different healthcare services. This will further help guide policy design and allocation of healthcare sources to alleviate both the cost and illness burden of sepsis on health system as well as patients.

There are some limitations that should be considered when interpreting these results. The cost estimates presented in this study should be interpreted as associations rather than a causal impact of sepsis. They do, however, offer a measure of the economic burden of sepsis care in cancer patients across 5 years of diagnosis which has not been previously quantified. The presence of sepsis could be confounded by a number of factors such as cancer stage or grade at diagnosis, treatments and comorbidities. We had not incorporated weighing methods in our case-control sample design which could have improved our selection of controls for the study. Although we have attempted to match for age, sex, cancer type and year of cancer diagnosis, our analysis was limited by the lack of complete information on these potential confounders to allow for adequate matching. It is possible that patients with sepsis had a late cancer stage at diagnosis, were on more aggressive treatments and/or had existing comorbidities which may predispose sepsis cases to incur higher costs [51, 58]. This could result in an over-estimation of the excess cost of sepsis. It may also be likely that among patients who developed sepsis, planned treatment programs may have been disrupted which can have variable cost implications. Further investigation to understand the impact of sepsis on patients at different cancer stages and its potential spill over impacts on treatment pathways, outcomes and associated costs is warranted. Additionally, large variations in survival and costs have been observed across different cancer types [29, 43], and an exploration of the burden of sepsis to reflect this heterogeneity will also be important. In exploring this, future costing analyses should also consider the usefulness of other statistical methods such as generalised linear models or two-part models that account for the unique properties of cost data and their applicability to specific research objectives [59].

## Conclusion

In summary, this study has demonstrated the substantial economic burden of sepsis in cancer patients over a 5-year period from initial cancer diagnosis using real-world population-linked data for a large cohort of cancer patients. Key efforts in improving sepsis prevention, recognition and management needs to be focused in the first year of cancer diagnosis when mortality and costs are highest. Given the increased susceptibility of this high-risk population to sepsis, these cost estimates are helpful in improving our understanding of burden of sepsis along the cancer pathway and to deploy targeted strategies to alleviate this burden. There should also be continued efforts in refining these estimates to reflect the heterogeneity across different cancer types.

## Supporting information

**S1 Appendix. Data source.**
(DOCX)

**S2 Appendix. Diagnostic codes used for identification of sepsis.**
(DOCX)

**S3 Appendix. Distribution of sepsis episodes from time of cancer diagnosis.**
(DOCX)

**S4 Appendix. Kaplan-Meier survival curves comparing sepsis cases to no sepsis controls.**
(DOCX)

**S5 Appendix. Descriptive statistics of sepsis cases vs. matched controls by malignancy types.**
(DOCX)

**S6 Appendix. Descriptive statistics of sepsis cases vs. unmatched cases by malignancy types.**
(DOCX)

**S7 Appendix. Breakdown of excess cost of care due to sepsis.**
(DOCX)

**S8 Appendix. Sub-group analyses results.**
(DOCX)

**S9 Appendix. Sensitivity analyses.**
(DOCX)

## Acknowledgments

This study was supported by ICES, which is funded by an annual grant from the Ontario Ministry of Health and Long-Term Care. The opinions, results and conclusions reported in this paper are those of the authors and are independent from the funding sources.

## Author Contributions

**Conceptualization:** Michelle Tew, Kim Dalziel, Karin Thursky, Philip Clarke.

**Formal analysis:** Michelle Tew.

**Methodology:** Michelle Tew, Kim Dalziel, Philip Clarke.

**Supervision:** Kim Dalziel, Karin Thursky, Murray Krahn, Lusine Abrahamyan, Andrew M. Morris, Philip Clarke.

**Writing – original draft:** Michelle Tew.

**Writing – review & editing:** Michelle Tew, Kim Dalziel, Karin Thursky, Murray Krahn, Lusine Abrahamyan, Andrew M. Morris, Philip Clarke.

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
