## [Decision Letter · Decision Letter 0]

9 Mar 2021

PONE-D-20-36215

High excess cost of care associated with sepsis in first year of cancer diagnosis: Results from a population-based case-control matched cohort

PLOS ONE

Dear Dr. Tew

Thank you for submitting your manuscript to PLOS ONE. After careful consideration, we feel that it has merit but does not fully meet PLOS ONE’s publication criteria as it currently stands. Therefore, we invite you to submit a revised version of the manuscript that addresses the points raised during the review process.

We look forward to receiving your revised manuscript.

Kind regards,

Edris Hasanpoor

Academic Editor

PLOS ONE

Journal Requirements:

Reviewers' comments:

Reviewer's Responses to Questions

**Comments to the Author**

1. Is the manuscript technically sound, and do the data support the conclusions?

Reviewer #1: Yes

Reviewer #2: Yes

Reviewer #3: Yes

Reviewer #4: Yes

Reviewer #5: Yes

2. Has the statistical analysis been performed appropriately and rigorously? 

Reviewer #1: Yes

Reviewer #2: Yes

Reviewer #3: Yes

Reviewer #4: I Don't Know

Reviewer #5: Yes

3. Have the authors made all data underlying the findings in their manuscript fully available?

Reviewer #1: Yes

Reviewer #2: Yes

Reviewer #3: No

Reviewer #4: No

Reviewer #5: Yes

4. Is the manuscript presented in an intelligible fashion and written in standard English?

Reviewer #1: Yes

Reviewer #2: Yes

Reviewer #3: Yes

Reviewer #4: Yes

Reviewer #5: Yes

5. Review Comments to the Author

Reviewer #1: This study, using administrative data, evaluates costs associated with sepsis in cancer patients. The analysis offers an intersting estimation of the high excess cost of care associated with sepsis in cancer patients comparing costs with a matched control group (cancer and no sepsis). The manuscript appears to be clear and well written. I suggest only minor revisions mainly in the discussion section:

1. Please expand on how the lack of information on the severity of the diseases might impact on final results;

2 Secopnd sentence in the discussion section ( Our results indicate that compared to patients without sepsis...) is not appropriate. In fact this is out of the scope of the study

Reviewer #2: I appreciate the authors’ curiosity to investigate economic burden of sepsis associated with cancer diagnosis and treatment. The paper shows the ice-Berge of the problem because it stands on health care perspective. But, the paper needs clarity on the following points:

1. What is the cut-off point to say high and low excess cost of care associated with sepsis?

2. When you were selecting samples, the weight of patient was not considered to select cases and controls, or not considering the weight of the patient was not putting as a part of study limitation. The weight of the patient could show the status of client and should be considered for case-control match sample selection. Otherwise, it is good to explain why the weight patient was not considered for case-control cross match sample selection.

3. Among Solid tumors, others account 42.3% which is greater than 3% which represents large groups of solid tumor cases, to be acceptable, it is good to put others into specific solid tumors with their percentage as long as it becomes less than 3%.

4. It might not be the objective of the study, but as depicted in the table 1, the cancer diagnosis was decreased in 2017 as compared to year before 2017. Do you have any justification?

5. The discussion part of the article discusses on indirectly related parameters to the objective of the study like satisfaction of sepsis survivors, and treatment options of cancer…. It is advisable to focus on our findings of the study. E.g. the paper discusses advanced treatment options but during the study treatment modalities were not considered.

Reviewer #3: Data is held within a system that will not be accessible to others who would perhaps like to replicate the study without going through a lengthy process and verification through the hosting organization (ICES). As per PLOS One policy, it would be best to anonymize the data utilized and upload onto a online server.

Table 2 could be transformed into a graph for better visualization of excess costs of cancer patients as compared to control.

Figure 1 & 2 is very low-resolution. Please double check.

Reviewer #4: This paper presents a retrospective cohort study of patients diagnosed with cancer who were further subdivided into two mutually samples of patients based on the type of cancer diagnosed (solid tumor vs. hematologic). Among these samples, healthcare costs of patients who developed sepsis were reported and compared against matched patients who did not develop sepsis. I think the paper is interesting, but believe the methodology should be described more explicitly and the authors’ conclusions need to be softened in light of the lack of causality between the excess costs observed and sepsis itself.

Generally speaking, the paper could benefit from a more thorough QC exercise. For example:

- Page 3, missing “the” before majority

- Page 3, “Robust cost estimates that provide long-term estimates” is redundant

- Page 4, sentence “This provides…” is difficult to read

- Page 6, “Methods that take (…) is required”, this is not conjugated properly

- Appendix 1, the footnotes are not numbered

- Appendix 4, haematology table, I think there is a p-value missing for type of cancer

- Appendix 5, Table A5, I think there is a p-value missing for the distribution of sexes

These examples may seem trivial, but they impede the effective reading of the manuscript. I encourage authors to QC their work more carefully prior to re-submitting this paper.

Introduction: One of the novel features of this study is the consideration of the global burden of sepsis beyond hospital costs. The authors point out that there are studies that looked at the inpatient costs of sepsis among cancer patients. It would be helpful to briefly describe these findings. Also, authors mention that these studies likely capture the most severe cases only. This needs further explaining – why are costs outside of the hospital context critical to the consideration of the burden of sepsis? What are the medications, allied health professionals services, rehabilitation services and follow-up outpatient visits that are not captured when looking at the inpatient costs only?

Methods: I have concerns about the way the authors have framed the research question. The authors claim in their objective to have “determine[d] healthcare costs associated with sepsis”. This wording implies a direct link between sepsis and the excess costs reported in the results. While I appreciate the effort to mitigate confounding through matching, patients who developed sepsis may have other underlying predispositions that make them more expensive on a 5-year time horizon. It would be more accurate to refer to the excess costs as “excess costs among patients who developed sepsis” rather than “costs associated with sepsis”. This change should be made throughout.

Patient cohort: Adult patients as of the cancer diagnosis, or the sepsis diagnosis? Please clarify why restrict to adults only, or re-frame the research questions to specify that this excludes paediatric patients.

Data source: “These data sources capture up to 90% of all healthcare resources provided” what is not included in these datasets?

I think the method of imputing costs based on the RIW and the unit cost of each service should be described more explicitly; the reader should be able to have an idea of the technique without having to refer to a citation. In addition to Appendix 1, an example would be very helpful so the reader can see how these RIW and unit costs play out in the estimations of patient costs. The concept of a “resource intensity weight” should be further described.

Patients were matched on cancer type. Table 1 shows a large proportion of “Others” for solid tumor patients. Were those all lumped together?

Another technique used by the authors is the imputation of costs among patients without complete follow-up. Since the time horizon is 5 years, and two thirds of patients were diagnosed on or after 2013, it appears that most of these patients do not have a full follow-up; this should be highlighted as a limitation and a sensitivity among patients with full follow-up should be considered. How are costs among remitted patients calculated, and how does the right censor affect the analysis of overall survival? And end-of-life care? Was the likelihood of death considered? Many things can happen to a patient in the 5 years that follow cancer diagnosis…

It would be interesting to present the distribution of time between cancer diagnosis and sepsis. The authors repeatedly claim that sepsis usually occurs shortly after the initiation of treatment, so the reader understands that costs associated with sepsis likely occur shortly following diagnosis. But if there is a material proportion of these patients for whom sepsis occurs years after the diagnosis of cancer, the conclusion does not stand. Also, Figures 1 start at month 0 but the sepsis could have occurred up to 30 days prior to diagnosis. Among patients whose sepsis was prior to cancer (n, %?), the cost of sepsis would be underestimated.

The authors bootstrapped cost estimates with a 1,000 replicates. This allows to center the confidence interval around the mean, but probably does not describe the cost distributions as they are in the cohort. The authors should explore using two-part modelling (likelihood of non-zero costs * multivariable regression with a gamma distribution) or GEE (cluster effect of matching) to estimate the difference between cohorts. These regression models would also help to introduce more control variables in the estimation of the cost difference. If the authors choose to uphold their current methodology, an explanation of the appropriateness of the method needs to be provided.

The authors appropriately acknowledged that cancer stage and comorbidities were not accounted for in the matching patients. The datasets used, however, seem to provide a very complete overview of the diagnoses, medications and services received by these patients, so I’m not sure why these were “not available” as per the Limitations. If they only received a data cut from ICES, why were those variables not requested in the data pull?

Furthermore, a sensitivity analysis by cancer type appears to be feasible based on the data available, and would help to provide additional context around the findings. Lung cancer and prostate cancer are two very different types of cancer with deferring prognosis and costs of treatment. It is likely that the likelihood of sepsis, and its associated incremental costs, would be different.

In the Discussion, the authors mention that the incremental costs of sepsis occurred shortly after diagnosis, which likely indicates that most episodes of sepsis occur shortly after diagnosis. I think this can be validated by comparing the costs of patients whose sepsis is within 30 days of diagnosis versus the others. Once again, having the distribution of time between cancer and sepsis would help the reader ascertain this conclusion. Also, the costs (and differences) taper over time. The authors hypothesize that 1) treatment following an episode of sepsis could be throttled back, or 2) there is a systemic lack of support among sepsis patients following their episode of sepsis. Do the authors believe that this tapering post-diagnosis is unlikely? I think the more plausible explanation is that sepsis is treated episodically, most costs occur within the inpatient setting, and the further out you are from diagnosis, the less expensive the cohorts are (patients' cancer could be cured, sepsis is over, patients could decease, etc.)

I was puzzled by some of the statistics mentioned in the Discussion. Authors mention that cost of care associated with sepsis increased by 85% for solid tumor and 179% for haematology patients. I’m not sure how the calculation was made: among solid tumor patients, incremental costs at year 5 are (60,714/72,969) 83% and among solid tumor patients (46,154/35,162) 90%. I could be doing the math wrong, but perhaps the sentence should be clarified then.

The conversation on sepsis pathways needs to be expanded.

Reviewer #5: This is an interesting paper and one which would be a useful addition to the literature. However, in its current format there are limitations which should ideally be rectified before publication.

The authors adopt a 1:1 matching approach using "age (+/-2 years), sex, cancer type, year of cancer diagnosis and rurality". These seem a sensible but limited set of variables to conduct matching. Why did the authors not also control for a wider set of variables (e.g. co-morbidities) which could influence the use of hospital resources above and beyond cancer and sepsis? Is it possible to remedy this limitation? It would also have been nice to see how the matching process altered the demographic profile of the matched controls within the study. This could be done by comparing the sepsis-negative group before and after matching.

The usability of the results is very much dependent on how the various resources are incurred. Unfortunately only an overall difference in cost is provided. It would be far more interesting and useful to see how this cost manifests itself into different resource-use categories. Is it a longer hospital stay or more expensive medication use in the sepsis cohort? Without further information, the reader learns that sepsis increases the cost of care for cancer which hardly seems revelatory. For those working in different countries it would be useful to see exactly where the extra costs fall. If possible, please report resource-use and costs separately (including the unit costs used as a supplementary appendix). Finally, the figures published within the main manuscript seem underwhelming. Information on wider resource-use would make a useful source to improve this aspect of the paper.

6. PLOS authors have the option to publish the peer review history of their article (what does this mean?). If published, this will include your full peer review and any attached files.

Reviewer #1: No

Reviewer #2: **Yes: **Abebe Ayinalem Tarekegn

Reviewer #3: No

Reviewer #4: No

Reviewer #5: No

---

## [Author Response · Author response to Decision Letter 0]

19 May 2021

Included in our submission is a detailed response to the reviewers’ comments.

---

## [Editor Report · Decision Letter 1]

28 Jun 2021

PONE-D-20-36215R1

Excess cost of care associated with sepsis in cancer patients: Results from a population-based case-control matched cohort

PLOS ONE

Dear Dr. Tew

Thank you for submitting your manuscript to PLOS ONE. After careful consideration, we feel that it has merit but does not fully meet PLOS ONE’s publication criteria as it currently stands. Therefore, we invite you to submit a revised version of the manuscript that addresses the points raised during the review process.

We look forward to receiving your revised manuscript.

Kind regards,

Edris Hasanpoor

Academic Editor

PLOS ONE

Additional Editor Comments (if provided):

Reviewer 1

This study, using administrative data, evaluates costs associated with sepsis in cancer patients. The analysis offers an intersting estimation of the high excess cost of care associated with sepsis in cancer patients comparing costs with a matched control group (cancer and no sepsis). The manuscript appears to be clear and well written. I suggest only minor revisions mainly in the discussion section:

1. Please expand on how the lack of information on the severity of the diseases might impact on final results;

2 Secopnd sentence in the discussion section ( Our results indicate that compared to patients without sepsis...) is not appropriate. In fact this is out of the scope of the study

Reviewer 2

I appreciate the authors’ curiosity to investigate economic burden of sepsis associated with cancer diagnosis and treatment. The paper shows the ice-Berge of the problem because it stands on health care perspective. But, the paper needs clarity on the following points:

1. What is the cut-off point to say high and low excess cost of care associated with sepsis?

2. When you were selecting samples, the weight of patient was not considered to select cases and controls, or not considering the weight of the patient was not putting as a part of study limitation. The weight of the patient could show the status of client and should be considered for case-control match sample selection. Otherwise, it is good to explain why the weight patient was not considered for case-control cross match sample selection.

3. Among Solid tumors, others account 42.3% which is greater than 3% which represents large groups of solid tumor cases, to be acceptable, it is good to put others into specific solid tumors with their percentage as long as it becomes less than 3%.

4. It might not be the objective of the study, but as depicted in the table 1, the cancer diagnosis was decreased in 2017 as compared to year before 2017. Do you have any justification?

5. The discussion part of the article discusses on indirectly related parameters to the objective of the study like satisfaction of sepsis survivors, and treatment options of cancer…. It is advisable to focus on our findings of the study. E.g. the paper discusses advanced treatment options but during the study treatment modalities were not considered.

Reviewer 3

Data is held within a system that will not be accessible to others who would perhaps like to replicate the study without going through a lengthy process and verification through the hosting organization (ICES). As per PLOS One policy, it would be best to anonymize the data utilized and upload onto a online server.

Table 2 could be transformed into a graph for better visualization of excess costs of cancer patients as compared to control.

Figure 1 & 2 is very low-resolution. Please double check.

Reviewer 4

This paper presents a retrospective cohort study of patients diagnosed with cancer who were further subdivided into two mutually samples of patients based on the type of cancer diagnosed (solid tumor vs. hematologic). Among these samples, healthcare costs of patients who developed sepsis were reported and compared against matched patients who did not develop sepsis. I think the paper is interesting, but believe the methodology should be described more explicitly and the authors’ conclusions need to be softened in light of the lack of causality between the excess costs observed and sepsis itself.

Generally speaking, the paper could benefit from a more thorough QC exercise. For example:

- Page 3, missing “the” before majority

- Page 3, “Robust cost estimates that provide long-term estimates” is redundant

- Page 4, sentence “This provides…” is difficult to read

- Page 6, “Methods that take (…) is required”, this is not conjugated properly

- Appendix 1, the footnotes are not numbered

- Appendix 4, haematology table, I think there is a p-value missing for type of cancer

- Appendix 5, Table A5, I think there is a p-value missing for the distribution of sexes

These examples may seem trivial, but they impede the effective reading of the manuscript. I encourage authors to QC their work more carefully prior to re-submitting this paper.

Introduction: One of the novel features of this study is the consideration of the global burden of sepsis beyond hospital costs. The authors point out that there are studies that looked at the inpatient costs of sepsis among cancer patients. It would be helpful to briefly describe these findings. Also, authors mention that these studies likely capture the most severe cases only. This needs further explaining – why are costs outside of the hospital context critical to the consideration of the burden of sepsis? What are the medications, allied health professionals services, rehabilitation services and follow-up outpatient visits that are not captured when looking at the inpatient costs only?

Methods: I have concerns about the way the authors have framed the research question. The authors claim in their objective to have “determine[d] healthcare costs associated with sepsis”. This wording implies a direct link between sepsis and the excess costs reported in the results. While I appreciate the effort to mitigate confounding through matching, patients who developed sepsis may have other underlying predispositions that make them more expensive on a 5-year time horizon. It would be more accurate to refer to the excess costs as “excess costs among patients who developed sepsis” rather than “costs associated with sepsis”. This change should be made throughout.

Patient cohort: Adult patients as of the cancer diagnosis, or the sepsis diagnosis? Please clarify why restrict to adults only, or re-frame the research questions to specify that this excludes paediatric patients.

Data source: “These data sources capture up to 90% of all healthcare resources provided” what is not included in these datasets?

I think the method of imputing costs based on the RIW and the unit cost of each service should be described more explicitly; the reader should be able to have an idea of the technique without having to refer to a citation. In addition to Appendix 1, an example would be very helpful so the reader can see how these RIW and unit costs play out in the estimations of patient costs. The concept of a “resource intensity weight” should be further described.

Patients were matched on cancer type. Table 1 shows a large proportion of “Others” for solid tumor patients. Were those all lumped together?

Another technique used by the authors is the imputation of costs among patients without complete follow-up. Since the time horizon is 5 years, and two thirds of patients were diagnosed on or after 2013, it appears that most of these patients do not have a full follow-up; this should be highlighted as a limitation and a sensitivity among patients with full follow-up should be considered. How are costs among remitted patients calculated, and how does the right censor affect the analysis of overall survival? And end-of-life care? Was the likelihood of death considered? Many things can happen to a patient in the 5 years that follow cancer diagnosis…

It would be interesting to present the distribution of time between cancer diagnosis and sepsis. The authors repeatedly claim that sepsis usually occurs shortly after the initiation of treatment, so the reader understands that costs associated with sepsis likely occur shortly following diagnosis. But if there is a material proportion of these patients for whom sepsis occurs years after the diagnosis of cancer, the conclusion does not stand. Also, Figures 1 start at month 0 but the sepsis could have occurred up to 30 days prior to diagnosis. Among patients whose sepsis was prior to cancer (n, %?), the cost of sepsis would be underestimated.

The authors bootstrapped cost estimates with a 1,000 replicates. This allows to center the confidence interval around the mean, but probably does not describe the cost distributions as they are in the cohort. The authors should explore using two-part modelling (likelihood of non-zero costs * multivariable regression with a gamma distribution) or GEE (cluster effect of matching) to estimate the difference between cohorts. These regression models would also help to introduce more control variables in the estimation of the cost difference. If the authors choose to uphold their current methodology, an explanation of the appropriateness of the method needs to be provided.

The authors appropriately acknowledged that cancer stage and comorbidities were not accounted for in the matching patients. The datasets used, however, seem to provide a very complete overview of the diagnoses, medications and services received by these patients, so I’m not sure why these were “not available” as per the Limitations. If they only received a data cut from ICES, why were those variables not requested in the data pull?

Furthermore, a sensitivity analysis by cancer type appears to be feasible based on the data available, and would help to provide additional context around the findings. Lung cancer and prostate cancer are two very different types of cancer with deferring prognosis and costs of treatment. It is likely that the likelihood of sepsis, and its associated incremental costs, would be different.

In the Discussion, the authors mention that the incremental costs of sepsis occurred shortly after diagnosis, which likely indicates that most episodes of sepsis occur shortly after diagnosis. I think this can be validated by comparing the costs of patients whose sepsis is within 30 days of diagnosis versus the others. Once again, having the distribution of time between cancer and sepsis would help the reader ascertain this conclusion. Also, the costs (and differences) taper over time. The authors hypothesize that 1) treatment following an episode of sepsis could be throttled back, or 2) there is a systemic lack of support among sepsis patients following their episode of sepsis. Do the authors believe that this tapering post-diagnosis is unlikely? I think the more plausible explanation is that sepsis is treated episodically, most costs occur within the inpatient setting, and the further out you are from diagnosis, the less expensive the cohorts are (patients' cancer could be cured, sepsis is over, patients could decease, etc.)

I was puzzled by some of the statistics mentioned in the Discussion. Authors mention that cost of care associated with sepsis increased by 85% for solid tumor and 179% for haematology patients. I’m not sure how the calculation was made: among solid tumor patients, incremental costs at year 5 are (60,714/72,969) 83% and among solid tumor patients (46,154/35,162) 90%. I could be doing the math wrong, but perhaps the sentence should be clarified then.

The conversation on sepsis pathways needs to be expanded.

Reviewer 5

This is an interesting paper and one which would be a useful addition to the literature. However, in its current format there are limitations which should ideally be rectified before publication.

The authors adopt a 1:1 matching approach using "age (+/-2 years), sex, cancer type, year of cancer diagnosis and rurality". These seem a sensible but limited set of variables to conduct matching. Why did the authors not also control for a wider set of variables (e.g. co-morbidities) which could influence the use of hospital resources above and beyond cancer and sepsis? Is it possible to remedy this limitation? It would also have been nice to see how the matching process altered the demographic profile of the matched controls within the study. This could be done by comparing the sepsis-negative group before and after matching.

The usability of the results is very much dependent on how the various resources are incurred. Unfortunately only an overall difference in cost is provided. It would be far more interesting and useful to see how this cost manifests itself into different resource-use categories. Is it a longer hospital stay or more expensive medication use in the sepsis cohort? Without further information, the reader learns that sepsis increases the cost of care for cancer which hardly seems revelatory. For those working in different countries it would be useful to see exactly where the extra costs fall. If possible, please report resource-use and costs separately (including the unit costs used as a supplementary appendix). Finally, the figures published within the main manuscript seem underwhelming. Information on wider resource-use would make a useful source to improve this aspect of the paper.

---

## [Author Response · Author response to Decision Letter 1]

7 Jul 2021

Response to Reviewers – Re Manuscript PONE-D-20-36215, “High excess cost of care associated with sepsis in first year of cancer diagnosis: Results from a population-based case-control matched cohort”

We are grateful to the Reviewers and Editor for their comments which we have responded to by amending the manuscript as set out below. Please see the attached Word document for our full response (which contains figures and tables to supplement our responses).

5. Review Comments to the Author

Reviewer #1: This study, using administrative data, evaluates costs associated with sepsis in cancer patients. The analysis offers an interesting estimation of the high excess cost of care associated with sepsis in cancer patients comparing costs with a matched control group (cancer and no sepsis). The manuscript appears to be clear and well written. I suggest only minor revisions mainly in the discussion section:

1. Please expand on how the lack of information on the severity of the diseases might impact on final results;

AUTHORS’ RESPONSE: We have now expanded our discussion to include potential impacts from differing disease characteristics. It now reads “The presence of sepsis could be confounded by a number of factors such as cancer stage or grade at diagnosis, treatments and comorbidities. Although we have attempted to match for cancer type and year of cancer diagnosis, our analysis was limited by the lack of complete information on these potential confounders to allow for adequate matching. It is possible that patients with sepsis had a late cancer stage at diagnosis, were on more aggressive treatments and/or had existing comorbidities which may predispose sepsis cases to incur higher costs [49, 50]. This could result in an over-estimation of the excess cost of sepsis. It may also be likely that among patients who developed sepsis, planned treatment programs may have been disrupted which can have variable cost implications. Further investigation to understand the impact of sepsis on patients at different cancer stages and its potential spill over impacts on treatment pathways, outcomes and associated costs is warranted.”.

2 Second sentence in the discussion section ( Our results indicate that compared to patients without sepsis...) is not appropriate. In fact this is out of the scope of the study

AUTHORS’ RESPONSE: We thank the reviewer for raising this as we realised we had not appropriately reflected where we drew our finding regarding mortality from. We have now removed this sentence to avoid deterring from the main focus of our paper which was on the cost burden of sepsis in cancer patients. As part of the costing exercise (described under Methods - Estimating Costs), survival probabilities for both cohorts were estimated and were used to adjust our cost estimates. The results from our survival analysis were described on page 8 (under Results – Study cohort and patient characteristics) and were presented in full in Appendix 4 in the Supplementary Materials.

Reviewer #2: I appreciate the authors’ curiosity to investigate economic burden of sepsis associated with cancer diagnosis and treatment. The paper shows the ice-Berge of the problem because it stands on health care perspective. But, the paper needs clarity on the following points:

1. What is the cut-off point to say high and low excess cost of care associated with sepsis?

AUTHORS’ RESPONSE: We thank the reviewer for the helpful comment. In this analysis, we had not defined a cut-off point for high or low excess cost. We described the excess cost to be high based on the results from the analysis which showed a large difference in cost between patients with sepsis and those without, particularly in the first 12 months of cancer diagnosis (Figure 1). For instance, the largest difference observed was in the first month of cancer diagnosis, and this was 179% higher for sepsis group (haematology) and 85% higher for sepsis group (solid tumour). However, in light of the reviewer’s comment, we have revised the title of our manuscript to avoid using the wording ‘high excess cost’ and amended our descriptions in the text of our manuscript to avoid misleading the readers. 

2. When you were selecting samples, the weight of patient was not considered to select cases and controls, or not considering the weight of the patient was not putting as a part of study limitation. The weight of the patient could show the status of client and should be considered for case-control match sample selection. Otherwise, it is good to explain why the weight patient was not considered for case-control cross match sample selection.

AUTHORS’ RESPONSE: The reviewer raises a good point and we had considered this. However, as we have drawn our controls (randomly selected) from the same population as the cases – from a large study base (Ontario Cancer Registry) which represented all reported cancer cases in Ontario, this ensured the selection was from a representative sample for the population of interest. In addition to this, we used a number of matching variables (age, sex, cancer type, etc.) to improve the efficiency of our matching [Wacholder et al (1992). Selection of controls in case-control studies: II. Types of controls. American journal of epidemiology, 135(9), 1029-1041]. The use of weight is most important in study designs where controls are drawn from complex stratified multi-stage studies or surveys, or selection of controls from sample that are likely to be a non-representative [Li et al (2011). Weighting methods for population‐based case–control studies with complex sampling. Journal of the Royal Statistical Society: Series C (Applied Statistics), 60(2), 165-185; Aigner et al (2018). Bias due to differential participation in case-control studies and review of available approaches for adjustment. PloS one, 13(1), e0191327; Scott et al (2009) Chapter 38 – Population-based case-control studies. Handbook of Statistics, 29(B), 431-453]. We reasoned that our study did not fall under this therefore had not incorporated the use of weights in our selection. We do acknowledge that due to the longitudinal nature of the data, we have patients that were observed over different time periods for the analysis therefore may contribute differently to the analysis. We overcame this by employing methods which reweights (adjusts) observed costs using survival probabilities (as described in Methods - Estimating Costs). This approach allowed for the estimation of unbiased cost estimates [Huang Y (2009) Cost analysis with censored data. Medical care. 47: S115; Wijeysundera et al (2012) Techniques for estimating health care costs with censored data: an overview for the health services researcher. ClinicoEconomics and outcomes research 4: 145].

Nonetheless, we have included a statement about not considering the weight of the patient as a limitation of our study in our revised manuscript. It now reads “We had not incorporated weighing methods in our case-control sample design which could have improved our selection of controls for the study. Although we have attempted to match for age, sex, cancer type and year of cancer diagnosis, our analysis was limited by the lack of complete information on these potential confounders to allow for adequate matching”.

3. Among Solid tumors, others account 42.3% which is greater than 3% which represents large groups of solid tumor cases, to be acceptable, it is good to put others into specific solid tumors with their percentage as long as it becomes less than 3%.

AUTHORS’ RESPONSE: Thank you for this suggestion. We have now expanded on our Others category as suggested by the reviewers. Changes have been made in Table 1. Table A4 in Appendix 5 further provides a more detailed breakdown of the different types of cancer. 

4. It might not be the objective of the study, but as depicted in the table 1, the cancer diagnosis was decreased in 2017 as compared to year before 2017. Do you have any justification?

AUTHORS’ RESPONSE: We reviewed our dataset containing all cancer diagnoses captured from 2010 to 2017 and the number of patients captured across the years remained relatively constant (see table below). All patients captured in the registry were followed up until death or end of the analysis period, March 31, 2018. As the reviewer has noted, there was a decrease in sepsis cases identified for the analysis in 2017 and this is likely to be due to the shorter follow up period available for the 2017 cohort. Although a large proportion of sepsis episodes occurred in the first year of cancer diagnosis, not all patients from the 2017 cohort had a full year of follow up and those developed sepsis beyond the follow up period would not be captured as cases. This likely explains the smaller proportion of cases from 2017. 

5. The discussion part of the article discusses on indirectly related parameters to the objective of the study like satisfaction of sepsis survivors, and treatment options of cancer…. It is advisable to focus on our findings of the study. E.g. the paper discusses advanced treatment options but during the study treatment modalities were not considered.

AUTHORS’ RESPONSE: We thank the reviewer’s advice and have refocused our discussion to reflect the findings directly observed from the analysis. We have removed the discussion points not directly relating to the study’s objectives. 

Reviewer #3: Data is held within a system that will not be accessible to others who would perhaps like to replicate the study without going through a lengthy process and verification through the hosting organization (ICES). As per PLOS One policy, it would be best to anonymize the data utilized and upload onto a online server.

AUTHORS’ RESPONSE: We thank the reviewer for raising this, and we agree that sharing anonymised data would be ideal. Unfortunately, our data use agreement with ICES prohibits us from sharing in this way even if anonymised. As the reviewer points out access may be granted to those who meet pre-specified criteria for confidential access, available at www.ices.on.ca/DAS. 

Table 2 could be transformed into a graph for better visualization of excess costs of cancer patients as compared to control.

AUTHORS’ RESPONSE: We agree that a graph would offer a better visualisation of the excess costs of cancer patients compared to control, however, we were more interested in displaying the actual values across different time frames such that these values can be used in cost-effectiveness models for decisions on sepsis interventions and are useful in helping inform development of sepsis programs and policies across the cancer care continuum. As such as we have retained Table 2. We have however, made some changes to Figure 1, which now shows the cost of care of cancer patients with sepsis and for those without sepsis, and shaded the region that represents the excess cost to visually show the magnitude. 

Figure 1 & 2 is very low-resolution. Please double check.

AUTHORS’ RESPONSE: We have improved the resolution of our figures. 

Reviewer #4: This paper presents a retrospective cohort study of patients diagnosed with cancer who were further subdivided into two mutually samples of patients based on the type of cancer diagnosed (solid tumor vs. hematologic). Among these samples, healthcare costs of patients who developed sepsis were reported and compared against matched patients who did not develop sepsis. I think the paper is interesting, but believe the methodology should be described more explicitly and the authors’ conclusions need to be softened in light of the lack of causality between the excess costs observed and sepsis itself.

Generally speaking, the paper could benefit from a more thorough QC exercise. For example:

- Page 3, missing “the” before majority

- Page 3, “Robust cost estimates that provide long-term estimates” is redundant

- Page 4, sentence “This provides…” is difficult to read

- Page 6, “Methods that take (…) is required”, this is not conjugated properly

- Appendix 1, the footnotes are not numbered

- Appendix 4, haematology table, I think there is a p-value missing for type of cancer

- Appendix 5, Table A5, I think there is a p-value missing for the distribution of sexes

These examples may seem trivial, but they impede the effective reading of the manuscript. I encourage authors to QC their work more carefully prior to re-submitting this paper.

AUTHORS’ RESPONSE: We thank the reviewer for these helpful comments, and we have taken steps to improve the quality of our manuscript during our revision. This includes the corrections changes suggested by the reviewer. We have also clarified aspects of the methodology raised by the reviewer according to the more specific suggestions below. 

Introduction: One of the novel features of this study is the consideration of the global burden of sepsis beyond hospital costs. The authors point out that there are studies that looked at the inpatient costs of sepsis among cancer patients. It would be helpful to briefly describe these findings. Also, authors mention that these studies likely capture the most severe cases only. This needs further explaining – why are costs outside of the hospital context critical to the consideration of the burden of sepsis? What are the medications, allied health professionals services, rehabilitation services and follow-up outpatient visits that are not captured when looking at the inpatient costs only?

AUTHORS’ RESPONSE: We have revised the Introduction section of our manuscript to include the points raised by the reviewer. The second paragraph in the Introduction now reads “Although sepsis incidence and its associated outcomes such as mortality have been well described in the literature [5-9, 13-16], the majority of these studies were focused on severe sepsis and were not specific to cancer. Among those that quantified costs, estimates [7, 14-17] have relied on hospital admissions data and showed that severe sepsis cancer hospitalisations can cost more than three times as much as non-severe cancer hospitalisations [14]. Hospitalisation data is likely to capture only the most severe cases and potentially miss sepsis burden incurred outside of the hospital. Previous studies have shown that the prevalence of less severe forms of sepsis is much higher than severe sepsis or septic shock, consequently the overall disease burden of sepsis is expected to be much larger [7, 18]. The burden of sepsis is also likely to extend beyond the index hospitalisation as growing evidence suggests that sepsis increases the risk of rehospitalisation [19, 20], cognitive decline [4, 21] cardiovascular complications [22, 23] and death [23-25] in studies assessing longer-term outcomes. Limited attention has focused on the economic burden of sepsis in the high-risk cancer population. Additionally, an understanding of the cost burden of sepsis beyond acute hospital care is needed to enable healthcare providers and policy makers to develop strategies for more efficient care. Currently, robust long-term cost estimates that adequately capture this in cancer populations are lacking.”. 

Methods: I have concerns about the way the authors have framed the research question. The authors claim in their objective to have “determine[d] healthcare costs associated with sepsis”. This wording implies a direct link between sepsis and the excess costs reported in the results. While I appreciate the effort to mitigate confounding through matching, patients who developed sepsis may have other underlying predispositions that make them more expensive on a 5-year time horizon. It would be more accurate to refer to the excess costs as “excess costs among patients who developed sepsis” rather than “costs associated with sepsis”. This change should be made throughout.

AUTHORS’ RESPONSE: We acknowledge the reviewers concerns on the need to clarify what the costs presented refer to. We have now revised our wording throughout our manuscript to reflect that suggested by the reviewer. We clarify in our limitation section that these cost estimates should not be interpreted as a causal impact of sepsis, and should be interpreted as an association instead. 

Patient cohort: Adult patients as of the cancer diagnosis, or the sepsis diagnosis? Please clarify why restrict to adults only, or re-frame the research questions to specify that this excludes paediatric patients.

AUTHORS’ RESPONSE: We have now revised the aim of our study to “In this study, we aim to describe short- and long-term healthcare costs of care of adult cancer patients with and without sepsis in Ontario, Canada.” and have also added the word adult patients to the first paragraph of our methods section to clarify this. 

Only adult patients (aged 18 and above whose first diagnosis for a primary cancer occurred between January 1, 2010 and December 31, 2017) identified from the Ontario Cancer Registry were included in our analysis. For patients below 18, complete demographic and treatment data may not be adequately captured in the datasets made available to us. A more reliable data source for paediatric patients is the Pediatric Oncology Group of Ontario Network Information System (POGONIS) which contains demographic, clinical and treatment data from Ontario’s five paediatric cancer centres, to which almost all Ontario children with cancer are referred. As we did not have access to this additional data source, we decided it was a better approach to limit our analysis to adult patients only, a population where we had more reliable data on. 

Data source: “These data sources capture up to 90% of all healthcare resources provided” what is not included in these datasets?

AUTHORS’ RESPONSE: Not captured in these datasets are the costs of healthcare services not paid by the Ontario Ministry of Health and Long-Term Care. This includes community services, outpatient prescriptions for those aged 65 and below (and not receiving social assistance) and other healthcare costs that are paid out-of-pocket [Wodchis et al. Guidelines on person-level costing using administrative databases in Ontario. Toronto: Health System Performance Research Network; 2011]. Despite this limitation, the data sources have been used in numerous costing analyses in Ontario [de Oliveira et al. Evaluation of Trends in the Cost of Initial Cancer Treatment in Ontario. CMAJ Open. 2013;1(4):E151–8; Krahn et al. Healthcare costs associated with prostate cancer: estimates from a population-based study. BJU Int. 2010;105(3):338–46; de Oliveira et al. The Costs of Cancer Care before and after Diagnosis for the 21 Most Common Cancers in Ontario. CMAJ Open. 2013;1(1):E1–8]. We have now added this additional description under Methods – Patient cohort and data source for clarification. It now reads “These data sources capture up to 90% of all healthcare resources provided universally and paid for by Ontario Ministry of Health and Long-Term Care [20]. Healthcare services and cost relating to community services, outpatient prescriptions for those aged 65 and below (and not receiving social assistance) and other healthcare costs paid out-of-pocket are not captured. Despite this, these data sources represent the best available and have been used in numerous other costing analyses [21-23].”

I think the method of imputing costs based on the RIW and the unit cost of each service should be described more explicitly; the reader should be able to have an idea of the technique without having to refer to a citation. In addition to Appendix 1, an example would be very helpful so the reader can see how these RIW and unit costs play out in the estimations of patient costs. The concept of a “resource intensity weight” should be further described.

AUTHORS’ RESPONSE: We have expanded our description of our costing approach in Appendix 1 with additional explanation of RIW and unit costs. We kept our description concise in the main manuscript to not distract from the main analysis and focus of the paper. Interested readers are directed to Appendix 1 which we have now revised to include a more detailed description. The approach we used to cost each of the health services presentations followed the comprehensive guidance provided by Wodchis et al, which describes in detail the justification and methodology for costing healthcare services using administrative datasets specific to Ontario. This resource is publicly available and widely used. 

Patients were matched on cancer type. Table 1 shows a large proportion of “Others” for solid tumor patients. Were those all lumped together?

AUTHORS’ RESPONSE: We have now expanded on our Others category as also suggested by Reviewer 2. Changes have been made in Table 1. A more detailed breakdown of the different cancer types is provided in Table A4 in Appendix 5. However, we acknowledge it may be more informative to provide more details in the main manuscript. 

Another technique used by the authors is the imputation of costs among patients without complete follow-up. Since the time horizon is 5 years, and two thirds of patients were diagnosed on or after 2013, it appears that most of these patients do not have a full follow-up; this should be highlighted as a limitation and a sensitivity among patients with full follow-up should be considered. How are costs among remitted patients calculated, and how does the right censor affect the analysis of overall survival? And end-of-life care? Was the likelihood of death considered? Many things can happen to a patient in the 5 years that follow cancer diagnosis…

AUTHORS’ RESPONSE: It appears that the reviewer may have misunderstood our approach in estimating costs. As such, we have taken steps to further clarify the description of our methodology in the revised manuscript. We had not imputed any costs for patients without complete follow-up. We described our data as right censored because we did not have complete follow-up for all patients until death or for the complete 5-year analytical period. This meant that for these patients, a portion of the relevant healthcare costs are unobserved because their observation period has ended prematurely. This is an issue commonly encountered with longitudinal data analysis. To overcome this issue and to ensure that all data appropriately contribute to the analysis, we employed a method that involved reweighting the average cost at each monthly interval by the probability of surviving to the start of next interval [Lin et al. (1997). Estimating medical costs from incomplete follow-up data. Biometrics, 419-434]. Although there are limitations to the current method; for example, the need to create time (monthly) intervals and assuming that censoring occurs at discrete monthly intervals, this remains a widely accepted approach to estimate costs. We have chosen this approach because (i) it compares well to other approaches such as the inverse probability weighted (IPW) estimator approach, particularly with smaller time intervals as we have employed in our analysis, and (ii) in the presence of heavy censoring, simple IPW methods may produce unstable estimates [Wijeysundera et al (2012) Techniques for estimating health care costs with censored data: an overview for the health services researcher. ClinicoEconomics and outcomes research 4: 145; O’Hagan et al. (2004). On estimators of medical costs with censored data. Journal of Health Economics, 23(3), 615-625].

As numerous studies have shown that cost of care generally increases towards time of death, we were also interested in capturing the proportion of costs that were attributed to costs nearing the last 6 months of life. As such, costs in the last 6 months of life were segmented into a separate category of ‘terminal care costs’ to distinguish these. We found that across the 5-year period, approximately 39% of the total excess cost was attributed to terminal care cost (last 6 months of life) in solid tumour patients. In haematology patients, the proportion of terminal care cost increased gradually over the 5-year period, from 36.8% at six months to above 90% by year 5. These terminal care cost estimates can be useful inputs for cost-effectiveness models that account for transitions cost to the death health state in the months prior to death when modelling sepsis interventions. 

Our methods section on Estimating costs on page 6 now includes further detail and clarification and reads “As patients were observed over different time periods, not all patients had complete cost information across the entire 5-year period. This meant that for these patients, a portion of the relevant healthcare costs was unobserved because their observation period ended prematurely (right censored). Therefore, to estimate costs with incomplete follow-up data (common in longitudinal studies), methods that take into account this form of censoring are required to ensure unbiased cost estimates [29, 30]. This was done by partitioning the study period into monthly intervals and adjusting observed costs at each interval by the survival probability of corresponding interval [31]. This approach was chosen because it compared well to other approaches such as the inverse probability weighted (IPW) estimator approach, particularly with smaller time intervals as we have employed in our analysis and in the presence of heavy censoring, simple IPW methods may produce unstable estimates [30, 32]. This provided estimates for mean monthly cost of care for cancer patients with sepsis (cases) and without sepsis (controls). The average total (cumulative) cost across 5 years was estimated as the sum across 60-monthly intervals. Excess (net) cost due to sepsis were estimated as the difference between the sepsis cases and no sepsis controls [33, 34]. As costs and survival probabilities are likely to be different between haematological and solid cancers, these patients were analysed separately. As cost of care at the end-of-life which is expected to be high [21, 34] and an important contributor to overall costs, costs in the last 6 months of life were segmented into a separate category of ‘terminal care costs’ to distinguish these. Sub-group analyses by sex and age groups were also conducted. Bootstrapping with 1000 replicates was used to calculate the 95% confidence intervals for all costs. All tests of significance used two-sided P-values at less than 0.05. Analyses were conducted using Stata version 16.”.

It would be interesting to present the distribution of time between cancer diagnosis and sepsis. The authors repeatedly claim that sepsis usually occurs shortly after the initiation of treatment, so the reader understands that costs associated with sepsis likely occur shortly following diagnosis. But if there is a material proportion of these patients for whom sepsis occurs years after the diagnosis of cancer, the conclusion does not stand. Also, Figures 1 start at month 0 but the sepsis could have occurred up to 30 days prior to diagnosis. Among patients whose sepsis was prior to cancer (n, %?), the cost of sepsis would be underestimated.

AUTHORS’ RESPONSE: We thank the reviewer for the suggestion. We agree with the reviewer that there are some patients who have developed sepsis years after cancer diagnosis and not all patients would necessarily have developed sepsis in the first year. 

Figures showing the proportion of cases that presented with sepsis at each monthly time period over 5 years. These figures have been added to the Appendix 3 in our Supplementary Materials and described in our results section “Across the 5-year period, a large proportion of sepsis episodes occurred in the first year of cancer diagnosis. Among haematology patients, 68.2% of first sepsis episodes were within the first year and this was 66.3% for solid tumour patients (Appendix 3). A higher proportion of haematology patients (41.0%) had >1 episode of sepsis compared to solid tumour patients (26.7%). The median time from cancer diagnosis to the first sepsis episode was 3 months (IQR, 0-12) for haematology patients and 4 months (IQR, 0-16) for solid tumour patients.”.

Our intention was to highlight that much of the cost burden of sepsis observed was within the first year of cancer diagnosis – where among haematology patients, 68.2% of first sepsis episodes were within the first year and this was 66.3% for solid tumour patients (shown in figures above). Patients who developed sepsis could also develop subsequent episodes of sepsis which will further add to the burden. We found that 41.0% of haematology patients and 26.7% of solid tumour patients had >1 episode of sepsis. Our findings are similar to that observed by Te Marvelde et al. who found rates of hospital admission for sepsis to be he highest in the year following cancer diagnosis and decreases in subsequent years [Te Marvelde et al (2020). Epidemiology of sepsis in cancer patients in Victoria, Australia: a population‐based study using linked data. Australian and New Zealand journal of public health, 44(1), 53-58]. 

In our sepsis case selection, we had included a 30-day period prior to cancer diagnosis to allow for some flexibility in accuracy in diagnosis dates and also inclusion of patients whose sepsis presentation may have been the result of undiagnosed cancer (following an approach from published literature [Te Marvelde et al. Epidemiology of sepsis in cancer patients in Victoria, Australia: a population‐based study using linked data. Australian and New Zealand Journal of Public Health. 2020; 44: 53-58]). This resulted in an additional 1,698 cases (2% of matched cases). We acknowledge the reviewer’s concerns regarding this and we have tested the impact of this in our sensitivity analysis (Appendix 9) by excluding the one-month pre-diagnosis from our sepsis case definition. The results for these additional analyses were presented in Table A12 (Appendix 9) which showed that exclusion of the 1-month pre-diagnosis period from our sepsis case definition produced cost estimates that were slightly higher (by 1 to 3%). Although our main analysis has slightly underestimated costs, this difference was small and it did not alter the conclusion that excess cost among cancer patients who developed sepsis was large. 

The authors bootstrapped cost estimates with a 1,000 replicates. This allows to center the confidence interval around the mean, but probably does not describe the cost distributions as they are in the cohort. The authors should explore using two-part modelling (likelihood of non-zero costs * multivariable regression with a gamma distribution) or GEE (cluster effect of matching) to estimate the difference between cohorts. These regression models would also help to introduce more control variables in the estimation of the cost difference. If the authors choose to uphold their current methodology, an explanation of the appropriateness of the method needs to be provided.

AUTHORS’ RESPONSE: We thank the reviewer for these suggestions and acknowledge that there are numerous ways to analyse costs – including econometric and regression models as proposed by the reviewer.

We have now added the following explanation in our methods section on Estimating costs “We employed the established “excess” or “net” cost approach to obtain costs attributable to sepsis, where estimated healthcare costs of cancer patients without sepsis was subtracted from the cost of cancer patients who developed sepsis [41, 42]. As it is often difficult to separate specific costs that are sepsis- or cancer-related, this approach has been applied in numerous costing analyses to describe economic burden associated with cancer [29, 30, 41-43].”. 

To further address this, we have also added in our limitation section “Additionally, large variations in survival and costs have been observed across different cancer types [29, 43], and an exploration of the burden of sepsis to reflect this heterogeneity will also be important. In exploring this, future costing analyses should also consider the usefulness of other statistical methods such as generalised linear models or two-part models that account for the unique properties of cost data and their applicability to specific research objectives [59].”.

For this particular research study, the objective was to describe the overall burden of sepsis across the cancer care continuum (from cancer diagnosis up to 5 years). With this in mind, we had approached our analysis using a population-based case-control study design and had used the established “excess” or “net” cost approach, where we estimated the total health care costs of cancer patients without sepsis and then subtracted this from the observed total cost of cancer patients who developed sepsis. All patients were matched at and followed up from the start of cancer diagnosis. Appropriate methodological considerations to account for different lengths of follow-up, and matching techniques were applied in the analysis. Costs were handled as described above to ensure unbiased cost estimates. The approach applied in our analysis reflected the type of study design and methodology that has been applied in high-impact publications that describe the economic burden associated with cancer. Some examples are as below:

• Schwartz, K. L., Simon, M. S., Bylsma, L. C., Ruterbusch, J. J., Beebe‐Dimmer, J. L., Schultz, N. M., ... & Quek, R. G. (2018). Clinical and economic burden associated with stage III to IV triple‐negative breast cancer: A SEER‐Medicare historical cohort study in elderly women in the United States. Cancer, 124(10), 2104-2114.

• Weycker, D., Akhras, K. S., Edelsberg, J., Angus, D. C., & Oster, G. (2003). Long-term mortality and medical care charges in patients with severe sepsis. Critical care medicine, 31(9), 2316-2323.

• Blakely, T., Atkinson, J., Kvizhinadze, G., Wilson, N., Davies, A., & Clarke, P. (2015). Patterns of cancer care costs in a country with detailed individual data. Medical care, 53(4), 302.

• de Oliveira, C., Pataky, R., Bremner, K. E., Rangrej, J., Chan, K. K., Cheung, W. Y., ... & Krahn, M. D. (2016). Phase-specific and lifetime costs of cancer care in Ontario, Canada. BMC cancer, 16(1), 1-12.

• de Oliveira, C., Weir, S., Rangrej, J., Krahn, M. D., Mittmann, N., Hoch, J. S., ... & Peacock, S. (2018). The economic burden of cancer care in Canada: a population-based cost study. CMAJ open, 6(1), E1. 

• Williams, M. D., Braun, L. A., Cooper, L. M., Johnston, J., Weiss, R. V., Qualy, R. L., & Linde-Zwirble, W. (2004). Hospitalized cancer patients with severe sepsis: analysis of incidence, mortality, and associated costs of care. Critical care, 8(5), 1-8.

The authors appropriately acknowledged that cancer stage and comorbidities were not accounted for in the matching patients. The datasets used, however, seem to provide a very complete overview of the diagnoses, medications and services received by these patients, so I’m not sure why these were “not available” as per the Limitations. If they only received a data cut from ICES, why were those variables not requested in the data pull? Furthermore, a sensitivity analysis by cancer type appears to be feasible based on the data available, and would help to provide additional context around the findings. Lung cancer and prostate cancer are two very different types of cancer with deferring prognosis and costs of treatment. It is likely that the likelihood of sepsis, and its associated incremental costs, would be different.

AUTHORS’ RESPONSE: Although the data sourced from ICES provided a comprehensive coverage of health services utilisation of cancer patients in Ontario, these data were largely extracted from administrative data, historically not commonly used for research purposes. Therefore, these data sources do not fully capture information on the clinical characteristics of patients. This unfortunately includes data on comorbidities. Although there are algorithms that are available which allows researchers to predict patient’s level of comorbidities from health resource utilisation (such as the John Hopkins Adjusted Clinical Groups System (https://www.hopkinsacg.org/), such resources were not freely and publicly available for use. 

Since 2010, substantial improvements have been made by Ontario Cancer Registry to include information regarding staging of cancer for selected cancers (prostate, breast, lung and colorectal). However, for the period of our analysis, this information remains incomplete to facilitate our matching process. This has therefore limited our analysis. As such we have acknowledged this as a limitation of our analysis and we hope with increased efforts to collect detailed staging information, future analyses will be able to segregate costs by stage as well.

The aim of the present analysis is focused on the quantifying the overall burden of sepsis in cancer patients as this has never been demonstrated before. These estimates will be of value in informing on health policy prioritisation. However, we agree with the reviewer that sepsis is likely to affect patients with different types of cancer differently (outcomes and associated costs of care) therefore, analyses by cancer types will be useful. We acknowledge that this is important and have plans to further our analyses to better understand the burden of care across the healthcare system and how this may vary across different cancer types. 

We have added the following points in our Discussion section “Further research to better understand pathways of care of cancer patients with sepsis is warranted. Enhancing our understanding of the role of different healthcare services can help guide policy design and allocation of healthcare sources to alleviate both the cost and illness burden of sepsis on health system as well as patients.” and “Further investigation to understand the impact of sepsis on patients at different cancer stages and its potential spill over impacts on treatment pathways, outcomes and associated costs is warranted. Additionally, large variations in survival and costs have been observed across different cancer types [29, 43], and an exploration of the burden of sepsis to reflect this heterogeneity will also be important.”. 

In the Discussion, the authors mention that the incremental costs of sepsis occurred shortly after diagnosis, which likely indicates that most episodes of sepsis occur shortly after diagnosis. I think this can be validated by comparing the costs of patients whose sepsis is within 30 days of diagnosis versus the others. Once again, having the distribution of time between cancer and sepsis would help the reader ascertain this conclusion. 

AUTHORS’ RESPONSE: Please see our response above to the reviewer’s comments regarding the distribution of time between cancer and sepsis. 

Also, the costs (and differences) taper over time. The authors hypothesize that 1) treatment following an episode of sepsis could be throttled back, or 2) there is a systemic lack of support among sepsis patients following their episode of sepsis. Do the authors believe that this tapering post-diagnosis is unlikely? I think the more plausible explanation is that sepsis is treated episodically, most costs occur within the inpatient setting, and the further out you are from diagnosis, the less expensive the cohorts are (patients' cancer could be cured, sepsis is over, patients could decease, etc.)

AUTHORS’ RESPONSE: We agree with the reviewer that an alternative explanation was that sepsis was treated episodically as the reviewer has suggested. We have revised our discussion to the following “We had anticipated excess cost of sepsis to remain substantial over the study period due to morbidities related to sepsis [4, 47, 48] and increased risk of sepsis in cancer survivors [49] which necessitates a greater level of care. However, we observed a long tail with much lower excess cost (Figure 1) over the 5-year time horizon. This could reflect the acute nature of sepsis, which is treated episodically, requiring intensive and expensive treatments when it occurs (likely within the inpatient setting where healthcare costs are high). Similar tapering trends in cost in the months following the initial diagnosis period have also been observed in previous studies similar that observed in the current study which may reflect the end of the intensive treatment and follow-up period [29, 43, 50]. It could also be due to a multitude of alternative factors; for instance, episodes of sepsis can lead to changes in the management of these patients including reduced intensity of treatments, cessation of therapy and/or prevention strategies for further episodes [51, 52].”.

I was puzzled by some of the statistics mentioned in the Discussion. Authors mention that cost of care associated with sepsis increased by 85% for solid tumor and 179% for haematology patients. I’m not sure how the calculation was made: among solid tumor patients, incremental costs at year 5 are (60,714/72,969) 83% and among solid tumor patients (46,154/35,162) 90%. I could be doing the math wrong, but perhaps the sentence should be clarified then.

AUTHORS’ RESPONSE: We have now clarified our statement. We had initially calculated this based on the difference observed between the two groups (sepsis vs. no sepsis) at each of the monthly intervals. And the largest difference observed across the 5 -year period was 179% higher for sepsis group (haematology) and 85% higher for sepsis group (solid tumour). It now reads “The cost of care of cancer patients who developed sepsis is substantial – up to 90% higher compared to patients without sepsis.”.

The conversation on sepsis pathways needs to be expanded.

AUTHORS’ RESPONSE: We have now expanded our discussion on this. The following text has been included in our discussion on sepsis pathways “Numerous intervention strategies and clinical pathways have been developed to assist clinicians in early identification of sepsis and prompt initiation of appropriate treatments to improve outcomes of patients with sepsis [45]. Clinical pathways for sepsis that provide protocolised management approaches such as sepsis bundles from the Surviving Sepsis Campaign have demonstrated effectiveness in reducing mortality by up to 50% [45-47]. However, majority of currently available literature are not specific to high-risk populations such as cancer. A clinical pathway with a whole-of-systems approach has the potential to alleviate the burden and costs of sepsis in cancer patients[17]. The findings from this analysis indicate the need to strengthen such initiatives for prompt sepsis identification and treatment particularly in the first year of cancer diagnosis and in managing (and preventing) subsequent sepsis episodes. Effective implementation of such strategies can have a big impact on improving outcomes of cancer patients with sepsis and in driving down the excess cost burden of sepsis as well as the future cost of managing sepsis and cancer.”.

Reviewer #5: This is an interesting paper and one which would be a useful addition to the literature. However, in its current format there are limitations which should ideally be rectified before publication.

The authors adopt a 1:1 matching approach using "age (+/-2 years), sex, cancer type, year of cancer diagnosis and rurality". These seem a sensible but limited set of variables to conduct matching. Why did the authors not also control for a wider set of variables (e.g. co-morbidities) which could influence the use of hospital resources above and beyond cancer and sepsis? Is it possible to remedy this limitation? It would also have been nice to see how the matching process altered the demographic profile of the matched controls within the study. This could be done by comparing the sepsis-negative group before and after matching.

AUTHORS’ RESPONSE: We thank the reviewer for raising this important point. As per our response to Reviewer 4 above, data used in this analysis were extracted from administrative data. Therefore, these data sources do not fully capture information on the clinical characteristics of patients, such as co-morbidities. Whilst we acknowledge this to be a limitation, we believe that our current analysis is still of value in providing estimates of the overall burden of sepsis in cancer patients. As per our response to Reviewer 3’ comments above, we have expanded on the limitations of our approach and discussed the potential implications in our Discussion section. 

We have provided a comparison of patients who were included in the analysis (sepsis cases) to those who were unmatched by malignancy types. The descriptive statistics of both groups of patients are presented in Tables A5 and A6 in Appendix 6. 

The usability of the results is very much dependent on how the various resources are incurred. Unfortunately only an overall difference in cost is provided. It would be far more interesting and useful to see how this cost manifests itself into different resource-use categories. Is it a longer hospital stay or more expensive medication use in the sepsis cohort? Without further information, the reader learns that sepsis increases the cost of care for cancer which hardly seems revelatory. For those working in different countries it would be useful to see exactly where the extra costs fall. If possible, please report resource-use and costs separately (including the unit costs used as a supplementary appendix). Finally, the figures published within the main manuscript seem underwhelming. Information on wider resource-use would make a useful source to improve this aspect of the paper.

AUTHORS’ RESPONSE: The reviewer makes an important and valid point about the need to further understand where the healthcare cost burden lies within the healthcare system. In the present analysis, we have focused on the quantifying the overall burden of sepsis in cancer patients as this has never been demonstrated before. These cost estimates can be helpful in informing on health policy prioritisation and also be used in cost-effectiveness models for decisions on sepsis interventions. We intend to further our analyses to better understand the burden of care across the healthcare system and how this may vary across different cancer types. This will involve analyses at a more granular level which will further improve our understanding on healthcare resource use and their associated costs. We appreciate the reviewer’s suggestions, and this will form part of a future research agenda.

---

## [Editor Report · Decision Letter 2]

12 Jul 2021

Excess cost of care associated with sepsis in cancer patients: Results from a population-based case-control matched cohort

PONE-D-20-36215R2

Dear Dr. Michelle Tew

We’re pleased to inform you that your manuscript has been judged scientifically suitable for publication and will be formally accepted for publication once it meets all outstanding technical requirements.

Kind regards,

Edris Hasanpoor

Academic Editor

PLOS ONE
---

## [Editor Report · Acceptance letter]

16 Jul 2021

PONE-D-20-36215R2 

Excess cost of care associated with sepsis in cancer patients: Results from a population-based case-control matched cohort 

Dear Dr. Tew:

I'm pleased to inform you that your manuscript has been deemed suitable for publication in PLOS ONE. Congratulations! Your manuscript is now with our production department. 

Kind regards, 

on behalf of

Dr. Edris Hasanpoor 

Academic Editor

PLOS ONE